# Comparative Genomic and Transcriptomic Profiling Revealed the Molecular Basis of Starch Promoting the Growth and Proliferation of *Balantioides coli*

**DOI:** 10.3390/ani13101608

**Published:** 2023-05-11

**Authors:** Lizhuo Zhao, Kai He, Chuanqi Jiang, Guangying Wang, Suhui Hu, Tianqi Wang, Weifeng Qian, Zhiguo Wei, Jie Xiong, Wei Miao, Wenchao Yan

**Affiliations:** 1Parasitology Laboratory, College of Animal Science and Technology, Henan University of Science and Technology, Luoyang 471023, China; 2National Animal Protozoa Laboratory, College of Veterinary Medicine, China Agricultural University, Beijing 100193, China; 3Key Laboratory of Aquatic Biodiversity and Conservation, Institute of Hydrobiology, Chinese Academy of Sciences, Wuhan 430072, China

**Keywords:** *Balantioides coli*, trophozoites, starch, transcriptomic analysis, comparative genomic analysis, proliferation, autophagy

## Abstract

**Simple Summary:**

*Balantioides coli* is a zoonotic parasite with a global distribution that inhabits the cecum and colon of humans and animals. This pathogen frequently causes diarrhea in weaned piglets undergoing the stress of transitioning from sow milk to pellet feed containing starch, leading to significant economic losses in the swine industry. Starch is necessary for the growth of the ciliated parasite. Here, our study revealed that *B. coli* is able to effectively consume and digest starch into glucose moieties. The molecular mechanism of the effect of starch on the growth and proliferation of *B. coli* by promoting the cell cycle and suppressing the autophagy of trophozoites has been elucidated. These findings will provide a novel insight to the pathogenesis of balantidial diarrhea in weaned piglets and an important reference for the control of balantiosis via adjusting the starch level in pellet feed.

**Abstract:**

Carbohydrates are the main source of nutrition for *B. coli*, supplying energy for cell growth and development. The research aimed at investigating the mechanism of starch on the growth and replication of *B. coli*. Single-cell separation was used to isolate single trophozoites of *B. coli* under a stereomicroscope, transcriptomic profiling was conducted based on the SMART-seq2 single-cell RNA-seq method. Comparative genomic analysis was performed on *B. coli* and eight other ciliates to obtain specific and expanded gene families of *B. coli*. GO and KEGG enrichment analysis were used to analyze the key genes of *B. coli* under the action of starch in the present study. The results of single-cell RNA-seq depicts starch affected the growth and replication of *B. coli* in two ways: (1) the cell cycle was positively promoted by the activation of the cAMP/PKA signaling pathway via glycolysis; (2) the cell autophagy was suppressed through the PI3K/AKT/mTOR pathway. Genes involved in endocytosis, carbohydrate utilization, and the cAMP/PKA signaling pathway were highly enriched in both specific and expanded gene families of *B. coli*. Starch can be ingested and hydrolyzed into glucose, in turn affecting various biological processes of *B. coli*. The molecular mechanism of the effect of starch on the growth and proliferation of *B. coli* by promoting cell cycle and inhibiting the autophagy of trophozoites has been elucidated in our study.

## 1. Introduction

*Balantioides coli* is an important zoonotic protozoan known to infect various hosts including humans, non-human primates, and pigs [1]. It is the largest protozoan known in humans, and pigs have been also proposed as the primary natural reservoir host for human infections [2]. The parasite is a food/water-borne pathogen with a direct life cycle, and is commonly transmitted by the fecal-oral route. Reportedly, in tropical and temperate regions, infections are most frequently found with a prevalence of 50–100% in pigs. *B. coli* can invade the colon and cecum of hosts and potentially cause intestinal diseases such as dysentery and even death [3], which is of great significance for animal husbandry and public health. The transmission of the parasite is actually influenced by several epidemiological factors including inadequate sanitation infrastructure (such as wastewater disposal and water supply), close contact with pigs, and poor hygiene. Both humans and animals are infected by the ingestion of cysts, and then cysts are converted to trophozoites under the action of digestive juices. Finally, trophozoites settle in the large intestine [4], where they are able to uptake and digest the starch in intestinal contents as their nutrients.

So far, studies of *B. coli* mainly focus on the diagnosis and treatment of clinical cases and epidemiological investigations, while the interaction of *B. coli* with its hosts, and its biochemistry and pathogenicity are not well understood. Barbosa et al. added sterile rice starch to the xenic medium Pavlova modified by Jones, and *B. coli* isolated from pigs and non-human primates were maintained in vitro for more than two years. The trophozoite died at the early stages of culture in media without starch [5]. In our previous study, a similar phenomenon was also observed in the modified DMEM culture medium. When a 5 mg/mL starch suspension was added to the DMEM medium, the activity of trophozoite stage of *B. coli* was increased, and the survival time was elongated [6]. These data implied that starch is necessary for the growth of *B. coli* and the parasite could use starch as its source of carbohydrates. Some previous investigations combined with our observations indicated that the prevalence and amount of *B. coli* in weaned piglets were higher than in other age groups [7]. More importantly, *B. coli* often cause diarrhea in weaned piglets when there is a sudden change of food from sow milk to pellet feed containing starch. Therefore, it is possible that the proportion of starch in feed will affect proliferation of the parasite in the large intestine of weaned piglets. However, the molecular basis of starch promoting the propagation of *B. coli* remains unclear.

Transcriptomic analysis is a powerful tool used to reveal metabolic pathways and relationships among various functional genes in cells or parasites, and to screen key functional genes under specific conditions [8,9]. In transcriptomic analysis of *Toxoplasma gondii*, a small RNA expressed by this parasite may be related to the response of the host immune cells and the adaptation of *T. gondii* to its host environment. In addition, some transcriptomic studies have been reported in *Cryptosporidium* [10,11]. To our knowledge, no associated transcriptomic research has been reported for *B. coli*. Recently, the single-cell RNA sequencing has gained substantial attention for its powerful ability to avoid cell heterogeneity when revealing gene expression and regulation in single cells. Here, both transcriptomic profiling based on single-cell RNA sequencing and comparative genomic analysis were performed to reveal the molecular mechanism of the effect of starch on the proliferation of *B. coli* in vitro.

## 2. Materials and Methods

### 2.1. Culture of B. coli In Vitro and Experimental Design

The *B. coli* strain P011 was originally isolated from a diarrheic fecal sample of the 43-day-old weaned piglet in Hongchang pig farm of Yichuan County, Henan Province used in this study, and the trophozoites were cultured in a modified DMEM medium at 28 °C as described previously. The detailed procedure and the composition of the modified DMEM culture medium were described previously [6]. Written informed consent was obtained from the owners of Hongchang pig farm for the participation of their piglets in this study.

A mouth pipette that is an aspirator tube assembly for a calibrated microcapillary pipette (Sigma-Aldrich Trading Co., Ltd., Shanghai, China) was used to isolate a single trophozoite of *B. coli* under a stereomicroscope [12,13]. The trophozoite sample at its peak of growth was taken to a clean culture dish and observed. The capillary tip of the mouth pipette was placed in the center of the field of vision, aiming at the moving trophozoite and inhaling the individual trophozoite into the capillary, then the trophozoite was blown into the ampoule with 3 mL of the basic DMEM medium. After manually picking one by one under the stereomicroscope, a total of 300 trophozoites were cultured in the modified DMEM containing 5 mg/mL starch (designated as the T group), while another 300 trophozoites were cultured in DMEM medium without starch as a negative control group (designated as the N group). The characteristics of growth and replication stages was characterized by the change of trophozoite concentration (parasites per milliliter), which in culture medium was counted by the limiting dilution method. A total of 100 μL of a well-mixed suspension of trophozoite culture was taken and diluted 10 times with the basic DMEM medium in a 2 mL centrifuge tube. After thoroughly mixing, 20 μL of diluted culture was taken and dropped onto a clean glass slide, and the number of trophozoites was counted using microscope with a 40× objective, which was defined as *B*. The trophozoite concentration (*A*) was calculated according to the following formula: *A* = *B* × 50 × 10 [6]. Three replicates of each group were performed to verify the reliability of the results.

For transcriptomic analysis, ten cells were isolated from trophozoite samples at the peak of growth as the initial sample (labeled as IS subgroup); at 72 h post starch addition, ten cells from T group as the starch treating sample (labeled as TS subgroup), and ten cells from N group as the negative control sample (labeled as NS subgroup). The isolated cells were washed 3–5 times in RNase-free water (Appendix A). Ten cells of each subgroup were mixed with 2 μL of cell lysis buffer (containing 0.2% (vol/vol) Triton X-100 and 2 U/μL RNase inhibitor). The released RNA in the RNase-free PCR tube (0.2 mL) was stored at −80 °C for single-cell RNA sequencing [14].

### 2.2. Construction of a cDNA Sequencing Library and PE150 Sequencing

The released RNAs were reverse-transcribed, template-switched, and amplified, and then cDNA libraries were constructed by modification of the SMART-seq2 single-cell RNA-seq method [14]. The cDNA was purified and recovered using Beckman Ampure XP magnetic beads and then dissolved in an elution buffer. The cDNA was measured by an Agilent 2100 High Sensitivity DNA Assay Kit (Agilent Technologies, Santa Clara, CA, USA) to assess the length, concentration, and quality of the amplified product. The amplified cDNA was used for the library construction of single-cell transcriptomes with an insert size of about 350 bp. The 150 bp paired-end reads were generated using Illumina HiSeq 4000 sequencing platforms for further analysis according to the standard protocols provided by ANOROAD gene Technology Co., Ltd. (Beijing, China).

### 2.3. Data Processing, Annotation, and Differential Expression Analysis

After sequencing, the raw data needed to be processed to obtain clean reads that could be used for subsequent analysis. The steps for filtering the raw sequencing data with software SeqPrep v 1.2 (https://github.com/jstjohn/SeqPrep, accessed on 5 August 2022) and Sickle v1.0 (https://github.com/najoshi/sickle, accessed on 5 August 2022) are as follows: (1) Filter out the adapter sequences in reads, and remove the reads that did not insert the fragments due to the adapter self-connection and other reasons. (2) Trim off the reads with N content more than 10% and the bases with low quality (mass value less than 30) at the end of the sequence (3’ end). (3) Discard adapters and sequences with less than 50 bp after trimming. After quality control, the statistical analysis and quality evaluation of the data were carried out again, and the clean reads were obtained.

The clean reads were aligned with the *B. coli* genome (GWHBOZN00000000, https://ngdc.cncb.ac.cn/gwh, accessed on 22 August 2022) using Hisat2 (v 2.2.1) to obtain mapped reads for subsequent analysis. The gene expression values calculated by FeatureCounts software (v 1.6.0) were subsequently standardized by reads per kilobase per million (RPKM). To reduce the effect of genes with low expression levels, the genes with a change of RPKM value below 10 were firstly filtered. Then, the differentially expressed genes (DEGs) within the three comparison groups were determined using the “edgeR” package (v 3.39.6) according to the criteria of fold change ≥ 2 and FDR < 0.05 [15]. The DEGs obtained in this study were labeled as TS vs. IS, NS vs. IS, and TS vs. NS sets.

In transcriptomic data analysis, clustering analysis is usually performed based on the number of genes to reflect the differences in gene expression between samples. Hence, in order to further explore the expression pattern among the three sets, DEGs of TS vs. IS, NS vs. IS, and TS vs. NS sets were combined for a hierarchical clustering analysis by the ‘pheatmap’ package of R software (v4.2.0).

### 2.4. Functional Enrichment and Metabolic Pathways Analysis

WEGO (http://wego.genomics.org.cn/, accessed on 30 August 2022) was first used for visualizing, comparing, and plotting Gene Ontology (GO) annotation results for DEGs from three sets, and the BiNGo plug-in of Cytoscape software v3.9.1 (https://cytoscape.org/, accessed on 10 September 2022) was used to perform GO enrichment analysis using the default parameters. The false discovery rate (FDR) was used to correct the statistical significance (*p* value) of each GO term. The protein sequences of all genes of *B. coli* were aligned to the Kyoto Encyclopedia of Genes and Genomes (KEGG) database (https://www.genome.jp/kegg/, release 89, accessed on 11 September 2022), and the KEGG Orthology (KO) results were obtained. According to the KO annotations corresponding to upregulated and downregulated genes, the metabolic pathways involved in the DEGs were searched by the KEGG Mapper tool (https://www.genome.jp/kegg/mapper/color.html, accessed on 11 September 2022).

### 2.5. Validation of DEGs by RT-qPCR

To verify the reliability of the data obtained by single cell RNA-seq, eight DEGs were selected for real time quantitative PCR (RT-qPCR) utilizing the primers listed in Appendix A. Primers of target genes were analyzed and designed using the software Oligo (v6.24). The total RNA was reverse-transcribed using an AT311-02 kit (TransGen-Biotech, Beijing, China). In a total volume of 20 µL, the qPCR mixture contained 0.4 μL of each primer (10 μM), 2 μL cDNA obtained from reverse transcription as template, 10 μL 2 × ChamQ Universal SYBR qPCR Master Mix (Vazyme#Q711), and 7.2 μL of nuclease-free water. The program for the RT-qPCR was performed with 7500 fast real-time quantitative PCR (BIO RAD, Hercules, CA, USA) for 45 cycles at 94 °C for 30 s, 94 °C for 15 s, 56 °C for 15 s, and at 72 °C for 20 s. The above RT-qPCR used the remaining cDNA samples of three biological replicates in each subgroup, and the differential expression was analyzed by the fold change method.

### 2.6. Comparative Genomic Analysis of B. coli and Eight Other Ciliates

The genome of *B. coli* (P011 strain isolated from a diarrheic fecal sample of the 43-day-old weaned piglet in our lab) was sequenced using the PacBio Sequel sequencing platform. The de novo assembly of *B. coli* genome was conducted using software and tools including Canu v2.0, WTDBG (https://github.com/ruanjue/wtdbg, accessed on 10 October 2022), Quickmerge v0.3 (https://github.com/mahulchak/quickmerge, accessed on 10 October 2022), Numer and Pilon v1.24 (https://github.com/broadinstitute/pilon, accessed on 10 October 2022), BUSCO v2.0 and CEGMA v2.5. The prediction and annotation of *B. coli* genome were completed by main tools and software including Augustus v2.4, GeMoMa v1.3.1, Hisat v2.0, TransDecoder v2.0, PASA v2.0, Infenal 1.1, BLAST v2.2, and Blast2GO. Finally, the *B. coli* genomic data have been obtained in the Parasitology Laboratory, Henan University of Science and Technology (GWHBOZN00000000, https://ngdc.Cncb.ac.cn/gwh, accessed on 10 March 2023).

The genomic data of eight closely related ciliate species (*Tetrahymena thermophila*, *Ichthyophthirius multifiliis*, *Paramecia tetraurelia*, *Stylonychia lemnae*, *Pseudocohnilembus persalinus*, *Oxytricha trifallax*, *Stentor coeruleus*, and *Halteria grandinella*) were downloaded and used for comparative genomic analysis of *B. coli*. Gene family clustering of nine ciliate species was performed with Orthofinder v2.4 to identify specific gene families of *B. coli* [16]. Gene family expansion and contraction analysis by café v4.2 (Computational Analysis of gene Family Evolution) were conducted to obtain the expanded gene families of *B. coli* [17]. Both specific and expanded gene families of *B. coli* were annotated with the PANTHER v15 database. Both GO and KEGG enrichment analyses were performed for specific and expanded gene families of *B. coli* using ClusterProfile v3.14.0.

### 2.7. Accession Numbers for Transcriptomic and Whole-Genome Sequence Data

The transcriptome sequencing data of *B. coli* reported in this paper have been deposited in the GSA, China National Center for Bioinformation/Beijing Institute of Genomics, Chinese Academy of Sciences, under accession number GSA008663 that is publicly accessible at https://ngdc.cncb.ac.cn/gsa/s/nd2L9ubV, accessed on 11 March 2023.

The whole-genome sequence data of *B. coli* reported in this paper have been deposited in the Genome Warehouse in the National Genomics Data Center, Beijing Institute of Genomics, Chinese Academy of Sciences/China National Center for Bioinformation, under accession number GWHBOZN00000000 that is publicly accessible at https://ngdc.cncb.ac.cn/gwh, accessed on 11 March 2023.

## 3. Results

### 3.1. Observation of the In Vitro Culture of B. coli

In the T group with 5 mg/mL starch, *B. coli* trophozoites engulfing starch particles could be observed under the microscope. The cilia enable the trophozoites to move rapidly and change direction at any time. The trophozoites displayed normal vitality (Appendix A). More importantly, the growth rate of trophozoites increased, as the trophozoite concentration reached a peak at 96 h post inoculation. However, in the N group without starch, the number of trophozoites gradually decreased, and trophozoites even disappeared at 96 h post inoculation. The trophozoites that lacked starch became smaller in size and weaker in vitality than in the T group. From 48 h to 72 h post inoculation, the number of trophozoites in the T group was significantly greater than in the N group (*p* < 0.05, Figure 1). These results highlighted the obvious effect of starch on the growth of *B. coli*.

### 3.2. Preliminary Analysis of Transcriptomic Sequencing Data

A total of 22.27 G clean data were generated in this study. As shown in Appendix A, the percentage of clean reads of the three subgroups IS, TS, and NS after filtering were 92.63%, 93.61%, and 87.32%, respectively. The proportions of Q20 for three subgroups reached 89.04−93.69%, and those of Q30 were more than 86.70%. In addition, GC content was 30% on average. The mapped percentages of transcriptome comparison were calculated as 68.2–76.1% in the three subgroups.

The results of correlation heatmap analysis based on the gene expression in the three subgroups are shown in Figure 2. The color difference between subgroups at different time points is clear, i.e., the correlations between the subgroups at different time points were low. With the extension of time, the overall change trends of TS and NS subgroups were similar; however, the gap between subgroups TS and NS gradually became larger, suggesting that starch had a large effect on the growth and proliferation of *B. coli*.

### 3.3. Differential Expression and Function Classification by GO Enrichment Analysis

To predict and analyze the classification and function of genes related to *B. coli* under the action of starch, the expression of DEGs among three comparison sets was determined. A total of 8876 DEGs were found in the NS vs. IS set, most of which were downregulated. In the TS vs. IS set, a total of 6493 DEGs were found in the TS subgroup with 5 mg/mL starch. For the TS vs. NS set, in contrast to the NS, 2231 downregulated genes and 5508 upregulated genes in the TS subgroup were identified to be involved in various metabolic changes (Figure 3A).

The GO annotation results of DEGs were displayed at GO level 3. In the NS vs. IS set, the numbers of downregulated genes in Cellular Component (CC), Molecular Function (MF), and Biological Process (BP) groups were greater than the respective numbers of upregulated genes (Appendix A), while in the TS vs. IS set, this phenomenon changed slightly, as the number of upregulated genes related to enzyme activity in CC was significantly higher than that of the set NS vs. IS (Appendix A). We noted that the results of the TS vs. NS set were completely different from those of the other two sets, and the number of upregulated genes was distinctly greater than the number of downregulated genes in most GO terms including the three categories CC, MF, and BP (Appendix A).

Meanwhile, the GO functional enrichment analysis of BP was further elucidated. The significantly upregulated genes under starch treatment were involved in processes such as protein post-translational modifications (phosphorylation), and microtubule-based movement in the TS vs. IS set (Figure 3B), while cellular protein metabolic process, regulation of transferase activity, protein kinase activity, and phosphorus metabolic process were found in the TS vs. NS set, accompanied by a series of basic cellular life processes (e.g., ribosome synthesis, protein synthesis, DNA replication, and the cell cycle) that were upregulated (Figure 3C).

### 3.4. Hierarchical Clustering Analysis of All Differentially Expressed Genes

To understand the dynamic changes of DEGs among the three subgroups, 11,451 genes in total were divided into four patterns (clusters 1–4) utilizing the hierarchical clustering method in the heatmap package. In Figure 4, each heatmap and dynamic line plot represent the corresponding change pattern of gene expression in individual cluster. Notably, in cluster 4, the gene expression level of TS was highest among the three subgroups, and that of NS was lower than IS, consistent with the pattern shown by the GO annotation results of TS vs. NS. GO enrichment analysis was performed for 3690 genes in cluster 4. The results showed that genes associated with calcium ion binding were enriched (Appendix A), an event that can enhance the catalytic activity of phosphorylase b kinase in glycogen metabolism, thereby enhancing glycolysis and powering the cells. In addition, compared with other subgroups, the DEGs of TS were enriched in the regulation of the cell cycle and phosphorylation (Appendix A), suggesting that they play a crucial role in cell division.

### 3.5. Analysis of KEGG Metabolic Pathways in B. coli under Starch Treatment

In accordance with the cluster analysis and GO enrichment analysis of DEGs, to determine the metabolic pathways of genes related to *B. coli* under the action of starch, the specific DEGs of the TS subgroup were mapped to the KEGG database. KEGG enrichment analysis results showed a significant alteration in the 48 pathways (Appendix A). In this study, we focused on the DEGs classified into five principal KEGG pathways, including starch and sucrose metabolism (Figure 5), glycolysis (Figure 6), PI3K-AKT signaling pathway (Appendix A), cell cycle (Appendix A), and mTOR signaling pathway (Appendix A). Compared with the NS subgroup, more genes related to the above biochemical pathways were upregulated in expression in the TS subgroup; the genes associated with metabolism of starch and sucrose as well as glycolysis were significantly upregulated, including AMY (K01176, EC:3.2.1.1), MA (K01208, EC:3.2.1.133), PYG (K00688, EC:2.4.1.1), and GPI (K01810, EC:5.3.1.9). Moreover, some genes related to cell proliferation such as Cdc25 (K03099), Cyr1 (K01768, EC:4.6.1.1), AKT (K04456, EC:2.7.11.1), SGK2 (K13303, EC:2.7.11.1), and CCNE (K06626) were also upregulated in expression. Meanwhile, the downregulated genes such as AMPK (K07198, EC:2.7.11.11) and ATG1 (K21357, EC:2.7.11.1) play vital roles in regulating cell autophagy (Table 1). Taken together, the results of the KEGG analysis confirmed that *B. coli* can hydrolyze starch to produce large amount of glucose, that can indirectly affect the level of cAMP, thereby regulating downstream biological processes, and ultimately affect cell autophagy, cell cycle, and cell proliferation.

### 3.6. Quantitative Real-Time PCR (RT-qPCR) Gene Expression Analysis

To verify the GO and KEGG enrichment analysis, eight key genes were chosen for further RT-qPCR testing (Figure 7). Those genes were distributed in hydrolysis of starch, glycolysis, cAMP signaling pathway, cell cycle, and autophagy of cells. The results showed that the DEGs obtained by RT-qPCR and RNA-seq analysis had a very high correlation (R^2^ = 0.86). Meanwhile, the comparative bar plot for corresponding genes expression of the two methods were shown in detail (Appendix A), the standard deviation was calculated form the three biological replicates. The results demonstrated that the data of differential gene expression of *B. coli* under the action of starch based on transcriptomic analysis were reliable in this experiment.

### 3.7. Specific and Expanded Gene Families of B. coli Associated with Intake and Utilization of Starch

Comparative genomic analysis of *B. coli* with other eight related species showed that 1436 specific gene families (including 4461 genes) and 36 expanded gene families (including 510 genes) were detected in the genome of *B. coli*. The expansion and contraction analysis of gene families among the nine species were estimated across a phylogenetic tree (Appendix A). The top ten enriched GO and KEGG terms of expanded gene families are included in Figure 8A and Appendix A. Genes related to endocytic vesicle and phagocytic vesicle in CC, catalytic activity such as hydrolase, in transporter activity, including glucose transporting in MF, and carbohydrate utilization in BP were significantly enriched in the GO analysis of *B. coli* expanded gene families. Genes involved in endocytosis, salivary secretion, pancreatic secretion, calcium signaling pathway, and cAMP signaling pathway were enriched in the KEGG analysis of expanded gene families.

Of the top ten enriched GO and KEGG terms of specific gene families for *B. coli*, genes related to endocytosis, ubiquitin mediated proteolysis, the MAPK signaling pathway and transmembrane transporter activities were enriched in the GO and KEGG analyses (Figure 8B and Appendix A). These genomic data supported the conclusion that *B. coli* trophozoites can consume and hydrolyze the starch particles into glucose by the regulation of these genes after adding starch to the culture medium.

## 4. Discussion

*B. coli* is an important common pathogen causing diarrhea in weaned piglets [18]. After weaning, the source of nutrition of piglets is changed from sow milk to pellet feed that contains starch. This sudden change in nutrition frequently induces balantidial diarrhea in weaned piglets, and this can cause large economic losses in the swine industry. In addition, starch is necessary for the growth and proliferation of *B. coli* in an in vitro culture system [5,6]. However, the molecular basis of the influence of starch on the growth and propagation of this parasite remains unclear. In the comparative genomic analysis of *B. coli* and eight other ciliates, some of the specific and expanded genes of *B. coli* related to the intake and hydrolyzation of starch were significantly enriched. The transcriptomic analysis based on single-cell RNA-Seq revealed the essential role of starch in the growth and proliferation of *B. coli* by promoting the cell cycle and inhibiting autophagy of cells as well. These findings will be very helpful for understanding the biochemistry and pathogenicity of *B. coli* in hosts.

In the in vitro culture of *B. coli*, the basic DMEM medium originally contained 4500 mg/L glucose, but the basic media used in the N and T group of this study were the same except for the presence of starch in the T group. In addition, the result of a previous experiment in the authors’ lab showed that the addition of glucose to the basic DMEM medium did not improve the growth and replication of *B. coli*, and the trophozoites died off after a period of time [6]. In GO enrichment analysis, compared to the IS subgroup, the trophozoites in the NS subgroup gradually depleted the stored starch in their body at a later stage of culture, the number of downregulated genes is more than that of the upregulated genes. More importantly, compared to the NS subgroup, the number of upregulated genes is significantly higher than that of downregulated genes in the TS subgroup. These data combined with some previous reports [5,6] suggest that starch plays a key role in the growth and replication of *B. coli*.

To develop a comprehensive overview of starch promoting the growth of *B. coli*, comparative genomic analysis was used to support the transcriptome results. In the comparative genomic analysis, genes involved in endocytosis were enriched most distinctly in the KEGG and GO enrichment analysis of specific and expanded gene families of *B. coli*. Our data support the hypothesis that trophozoites are able to intake starch particles in the culture medium or the large intestine of hosts through the oral apparatus located apically, and then the consumed starch particles at the bottom of the oral apparatus are invaginated into a food vacuole. These food vacuoles will be filled with small starch particles and then function as temporary storage of food including starch in the cytoplasm of *B. coli*. More importantly, genes related to salivary secretion, pancreatic secretion, transporter activity including glucose transport, and the cAMP/PKA signaling pathway were also significantly enriched in the KEGG and GO analysis of expanded gene families. As is well known, the salivary glands and pancreas of mammals are able to secrete amylases. The genes of alpha amylase (AMY) and other maltase were upregulated for expression in *B. coli* after starch treatment in this study. These findings imply that *B. coli* is able to produce a number of amylases and maltase for the hydrolyzation of starch into glucose, and then the generated glucose is transported to the right sites in the cytoplasm through glucose transporters. Our previous data suggest that *B. coli* may be a facultative anaerobic organism [6]. Therefore, each generated glucose will be decomposed into two pyruvic acid molecules and release two ATPs by the anaerobic glycolysis pathway. The resulting ATPs and intermediate products can provide energy and serve as a carbon source for the biosynthesis of proteins and other nutrients in *B. coli*.

In *Saccharomyces cerevisiae*, the cAMP/PKA signaling pathway is dually activated by glucose via glycolysis and the G-protein coupled receptor Gpr1 [16]. Glycolysis generates fructose-1,6-bis-phosphate (F-1,6-BP) that activates the Ras nucleotide exchange factor Cdc25 and in turn activates the GTPases Ras1 and Ras2. GTP-bound Ras1 and Ras2 promote cAMP production via activation of the adenylyl cyclase Cyr1. Gpr1 senses external glucose and is activated via the GTPase Gpa2, and then stimulates Cyr1 to produce cAMP. The cAMP promotes the dissociation of the catalytic (Tpk1, 2, 3) and regulatory (Bcy1) subunits of PKA, leading to the activation of PKA and downstream signaling, and finally resulting in cell proliferation of *S. cerevisiae* [19]. In *B. coli*, the Gpr1-like receptor may not have occurred on the membranes of trophozoites due to the lack of growth or even death in the DMEM medium containing 4500 mg/L glucose without starch after 48 h post inoculation. Conversely, the trophozoites were able to grow well in the modified DMEM culture medium with 5 mg/mL starch. Consequently, we speculated that trophozoites failed to use extracellular glucose directly, and the intake and hydrolysis of starch may be the only way to acquire glucose by *B. coli*. In this study, genes related to the glycolysis and cAMP/PKA pathways such as PYG, pfk, Cdc25, and Cyr1 were enriched and upregulated in expression in the trophozoites of *B. coli* after starch treatment. The data imply that the product of glycolysis in *B. coli*, F-1,6-BP can activate the cAMP/PKA pathway and in turn regulate downstream biological process including the cell cycle [17,20,21].

In the cell cycle of *Tetrahymena* and other protists, elevated cAMP can directly activate Epac that indirectly affects AKT and PDK1 via PI3K. The phosphorylation of AKT promotes the upregulation of Cyclin and CDK; the former plays an important part in cell mitosis, while the latter is a considerable factor in cell cycle regulation in eukaryotes [22,23,24]. A high level of PDK1 can activate SGK [25], and SGK can be stimulated by various growth factors to promote cell survival and inhibit cell apoptosis and suppress apoptosis [26]. In addition, phosphorylation of PDK1 promotes the upregulation of AKT, a factor that is significant for cell survival [27]. Here, genes associated with the cell cycle such as PDK1, AKT, CDK, Cyclin, and SGK were detected to be upregulated, while PI3K and Rap1 were downregulated in *B. coli* after starch treatment. Our data support the hypothesis that the cAMP promoted by F-1,6-BP from glycolysis can activate the cell cycle of *B. coli* trophozoites via the related regulatory factors.

cAMP depends on AKT in regulating autophagy. The PI3K/AKT/mTOR signaling pathway has been demonstrated to be the main pathway for cells to regulate autophagy [28,29,30,31]. mTORC1 is directly involved in the regulation of some major cellular autophagy proteins, and its upstream regulatory factor is PI3K/AKT, which can integrate signal pathways from the cellular environment and eventually regulate autophagy [32]. Studies have shown that phosphorylation of AMPK directly activates ATG1 in glucose deficiency, thereby promoting autophagy [33]. When nutrients are sufficient, the expression of AMPK is reduced, and its inhibitory effect on the phosphorylation of mTORC1 is weakened [34]. However, the activated mTORC1 by a high level of AKT disrupts the interaction between AMPK and ATG1, preventing the activation of ATG1, thereby suppressing autophagy [35]. In this study, PI3K, AMPK, and ATG1 genes regulating cell autophagy were downregulated, while AKT was upregulated in *B. coli* at 72 h post starch treatment. These regulatory factors were engaged in the suppression of autophagy of the trophozoites.

According to the data of transcriptomic analysis based on single-cell RNA sequencing of *B. coli* trophozoites after starch treatment combined with comparative genomic analysis, a hypothetical molecular mechanism model for the effect of starch on the growth of *B. coli* was constructed (Figure 9) as follows: The starch supplied into the culture medium is consumed by trophozoites, and then invaginated and hydrolyzed into glucose by α-amylase and maltase in the food vacuole. The glucose is transported out of food vacuoles to the proper sites in the cytoplasm by related glucose transporters, and finally anaerobic glycolysis occurs. The glycolysis generates ATP, pyruvic acid, and other intermediate products including F-1,6-BP. The F-1,6-BP is sensed by Cdc25, which in turn activates the GTPases Ras1 and Ras2. GTP-bound Ras1 and Ras2 promote cAMP production via activation of Cyr1. The elevated cAMP directly activates Epac that indirectly promotes AKT and PDK1 expression via Rap1 and PI3K. The phosphorylation of AKT promotes the upregulation of Cyclin and CDK. The phosphorylation of PDK1 activates SGK. The high levels of Cyclin, CDK, and SGK drive cell division and proliferation and thus complete the cell cycle. In addition, the phosphorylation of AKT with upregulated expression will indirectly activate mTORC1, and then mTORC1 can prevent ATG1 activation and disrupt the interaction between AMPK and ATG1, and thus suppresses autophagy of cells. Finally, trophozoites of *B. coli* can constantly propagate and grow in the culture medium containing the right starch level.

To our knowledge, this is the first report of single-cell transcriptomic study of *B. coli*. In fact, transcriptomic analysis still has certain limitations, including some of replicates in subgroups failing to be sequenced, and not directly proving whether the phenotype has changed, further multi-omics studies including transcriptomics and metabolomics with a greater sample size are needed to explain the complexity and integrity of biological processes of *B. coli.*

## 5. Conclusions

Taken together, the effective utilization of starch in *B. coli* was supported through comparative genomic analysis and transcriptomic profiling. The molecular basis of starch promoting the growth and proliferation of *B. coli* was revealed by activation of the cAMP/PKA signaling pathway via glycolysis, which positively promoted the cell cycle, and suppression of cell autophagy through the PI3K/AKT/mTOR pathway in our study. These findings will be helpful for us to understand the pathogenesis of balantidial diarrhea and provide an important reference for the control of balantiosis in weaned piglets and other young animals by adjusting the starch level in their feed. However, the functions of the key genes regulating starch utilization, glycolysis, and other related metabolic pathways will need to be further confirmed using other tools such as gene knock-out strategy.

## Figures and Tables

**Figure 1 animals-13-01608-f001:**
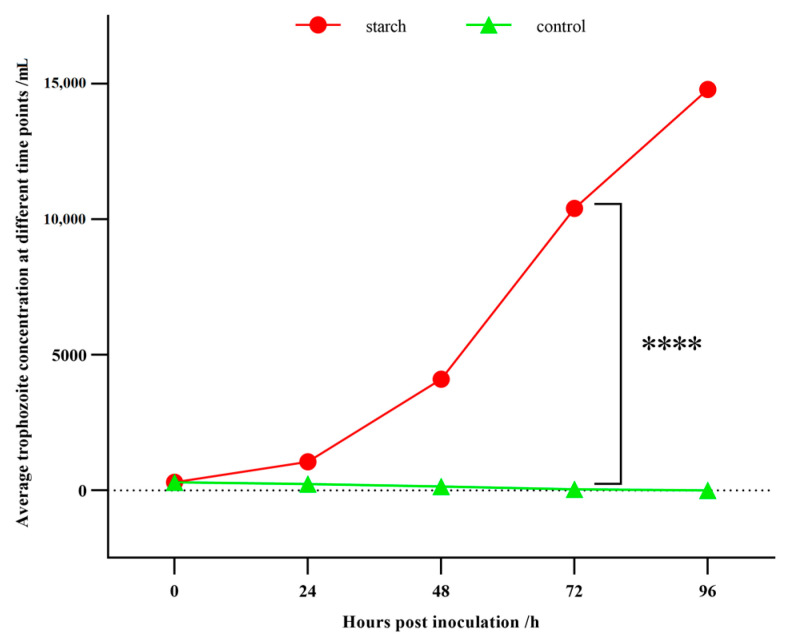
Statistics of the trophozoite concentrations of *B. coli* at various time points. The culture media of T (with starch) and N (without starch) groups were inoculated with 300 trophozoites. At 0, 24, 48, 72, 96 h post-inoculation, trophozoites were counted under a microscope using the limiting dilution method. Statistical analysis was performed using the one-way ANOVA and significant difference was observed between the T and N groups at 72 h (****: *p* < 0.0001). The concentration at 0 h post inoculation was 100 cells/mL, the average concentration at 72 h in T group was 10,400 cells/mL, and the average trophozoite concentration at 72 h in N group was 30 cells/mL. The average trophozoite concentration was calculated based on three biological replicates.

**Figure 2 animals-13-01608-f002:**
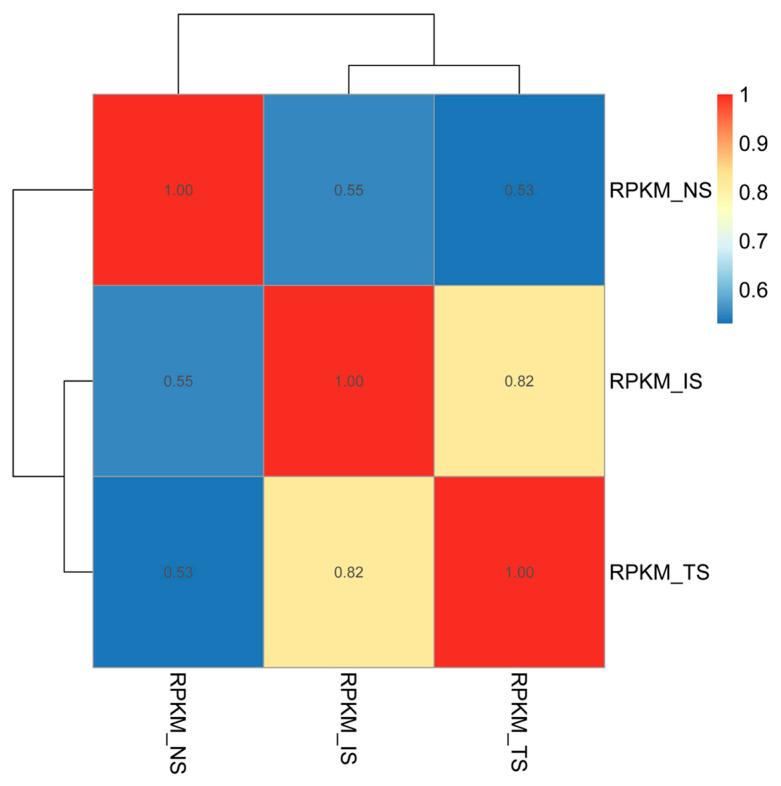
Pearson correlation heatmap of three subgroups. The Pearson correlation coefficients were calculated using the gene expression level between pairs of subgroups. The color closer to red denotes a strong correlation between two subgroups. The difference in gene expression levels was distinct between the TS and IS subgroups.

**Figure 3 animals-13-01608-f003:**
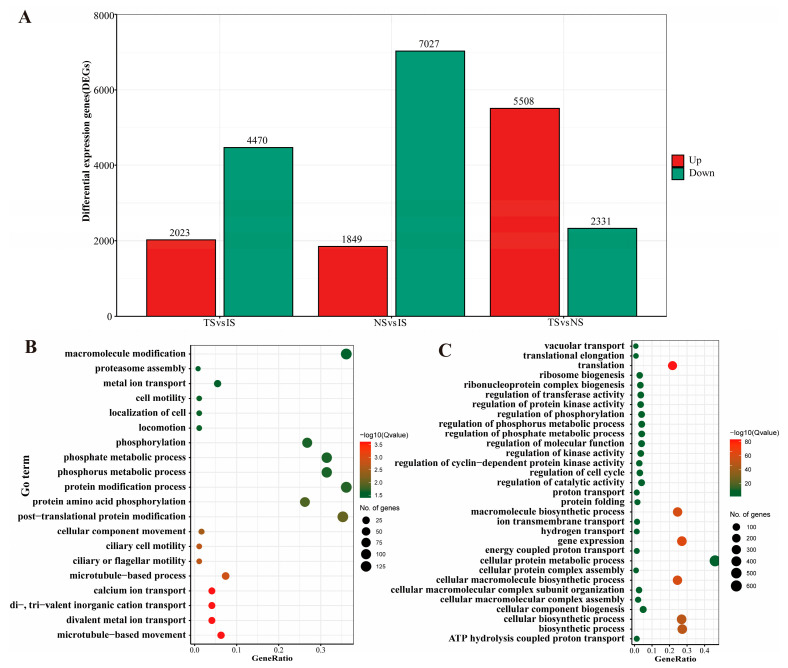
Statistics of differentially expressed genes and their GO functional enrichment analysis. (**A**): Statistics of differentially expressed genes among different comparison sets. (**B**,**C**): GO functional enrichment analysis of DEGs for the biological process (BP) in TS vs. IS and TS vs. NS sets, respectively. GeneRatio represents the ratio of the number of identified genes to the number of total genes annotated in the individual term.

**Figure 4 animals-13-01608-f004:**
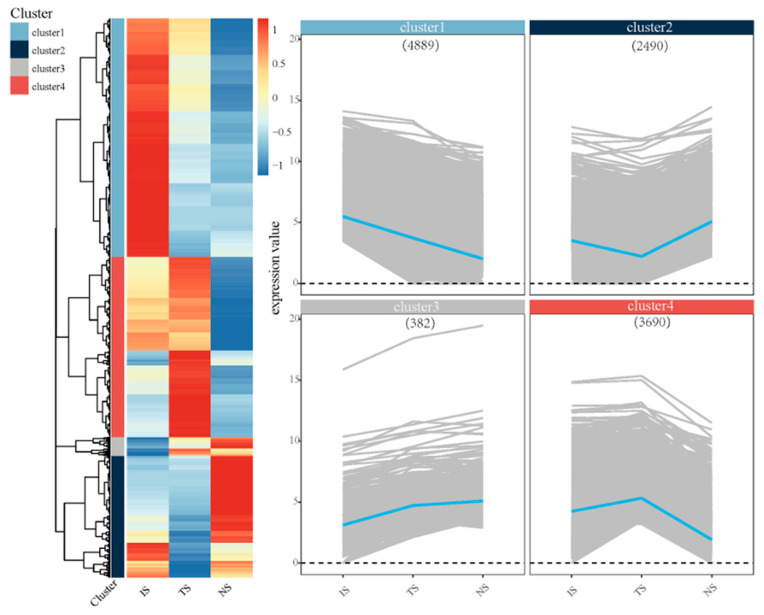
A hierarchically clustered heatmap for all identified DEGs in three subgroups. The gene clustering heatmap and trend diagram illustrate the different expression pattern, and expression levels of DEGs are shown as mean-centered log2 (RPKM + 1) values for the trend pattern. All DEGs from the three sets were divided into four clusters and labeled with different colors. In cluster 4, labeled red, the expression levels of 3690 genes in the TS subgroup were higher than in the other two subgroups.

**Figure 5 animals-13-01608-f005:**
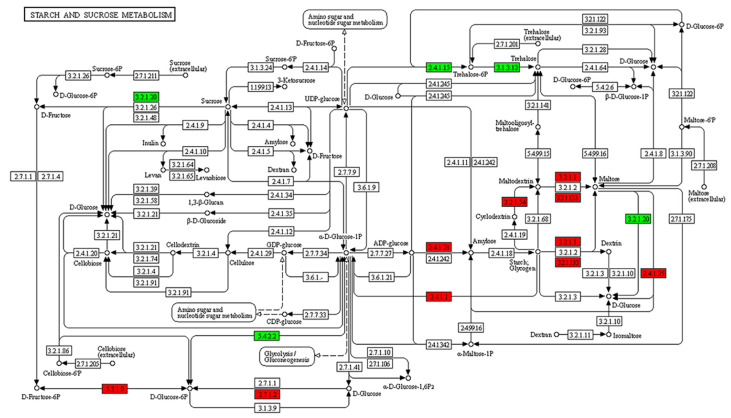
KEGG pathway analysis of starch and sucrose metabolism in *B. coli* based on the DEGs of the set TS vs. NS. The red and green colors represent the up- and downregulations of genes involved in the pathways of starch and sucrose metabolism, respectively. The genes related to alpha amylase, maltogenic alpha-amylase, and 4-alpha-glucanotransferase were upregulated in expression in the hydrolyzation of starch in *B. coli*.

**Figure 6 animals-13-01608-f006:**
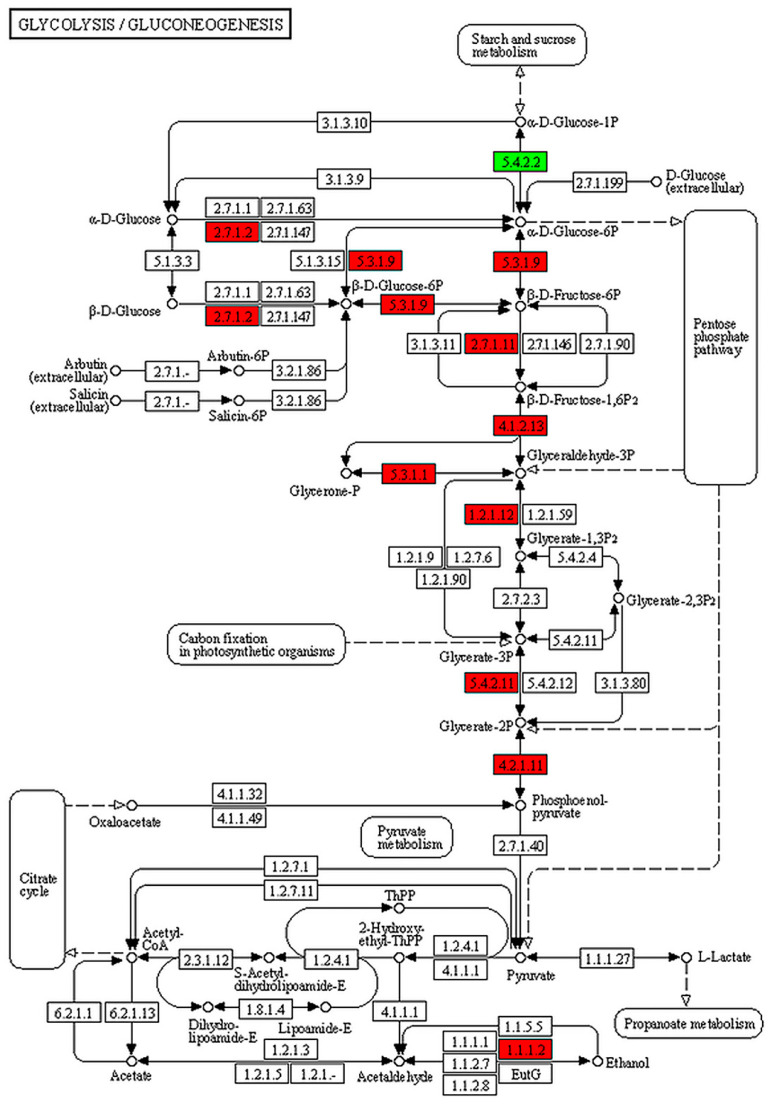
KEGG pathway analysis of glycolysis in *B. coli* based on the DEGs of the set TS vs. NS. The red and green colors represent the up- and downregulations of genes involved in the glycolysis pathway. The genes related to glucokinase, glycogen phosphorylase, glucose-6-phosphate isomerase, and 6-phosphofructokinase were upregulated in expression in the glycolysis pathway of *B. coli*.

**Figure 7 animals-13-01608-f007:**
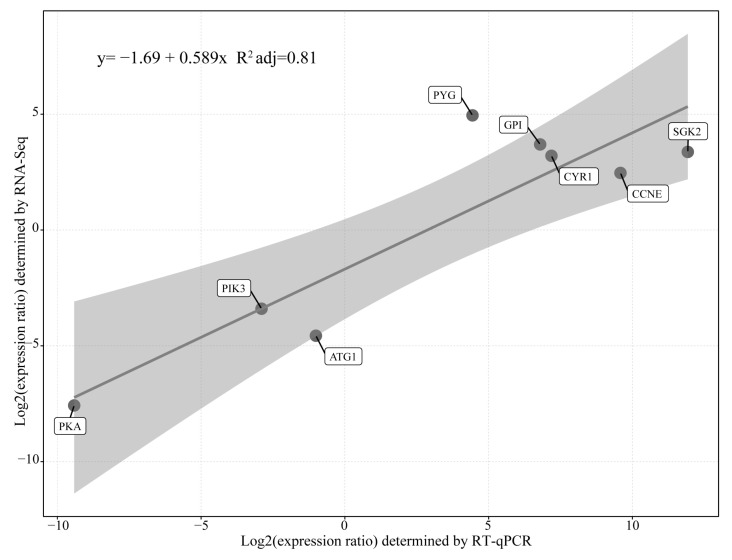
RT-qPCR confirmation of the single cell RNA-seq results. Confirmation of expression of eight genes based on Pearson correlations (r values) between FCs (log2 scale) reported by the RT-qPCR and single cell RNA-seq results.

**Figure 8 animals-13-01608-f008:**
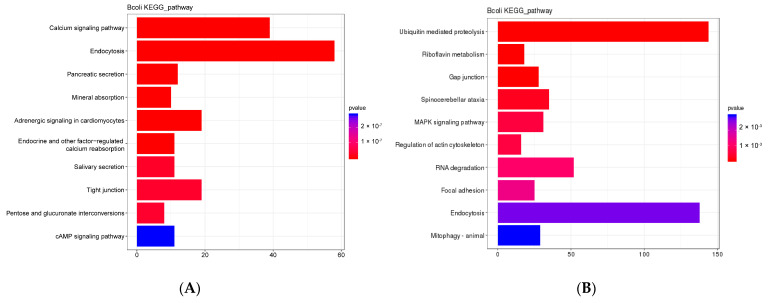
KEGG enrichment analysis of expanding (**A**) and specific (**B**) gene families of *B. coli*. The genes related to endocytosis, salivary secretion, pancreatic secretion, calcium signaling pathway, and the cAMP signaling pathway were significantly enriched in *B. coli*. In the histogram, the horizontal coordinate is the number of genes under this pathway, and the bar colors represent the corresponding *p* values; a lower *p* value refers to more significant enrichment.

**Figure 9 animals-13-01608-f009:**
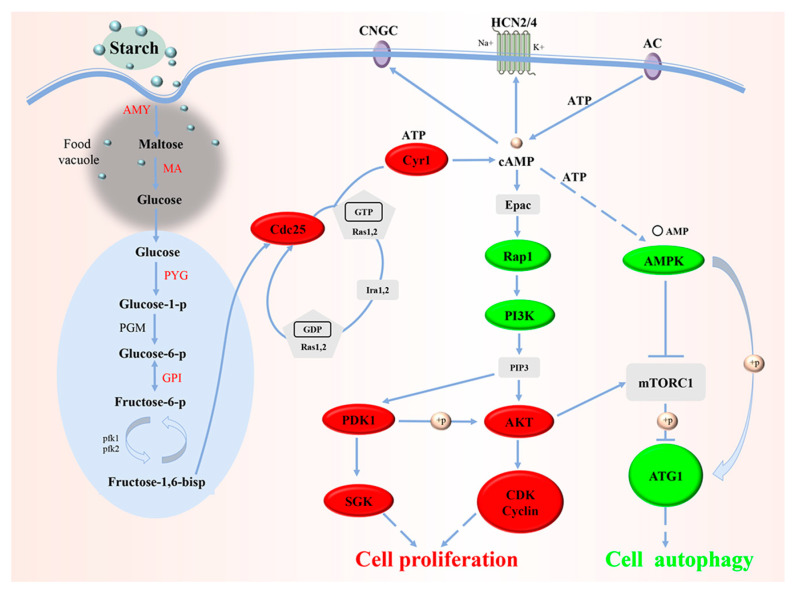
The hypothetical molecular mechanism model was developed for the influence of starch on the growth and proliferation of *B. coli*. This model includes starch hydrolysis, glycolysis, activation of the cAMP/PCA signaling pathway, the cell cycle, and autophagy of cells in *B. coli*. The red and green colors represent the genes with upregulated and downregulated expression in the related pathway, respectively.

**Table 1 animals-13-01608-t001:** The major differentially expressed genes related to hydrolyzation of starch, glycolysis, and the cell cycle and autophagy of cells in *B. coli* under the action of starch.

Symbols	Full-Length Names	EC Number	KO	TS Subgroup vs. NS Subgroup			
				Regulated	Log2 (FC)	*p* Value	Padjust
AMY ^a^	alpha-amylase	3.2.1.1	K01176	up	3.58	1.49 × 10^−9^	1.77 × 10^−8^
MA ^a^	maltogenic alpha-amylase	3.2.1.133/3.2.1.54	K01208	up	2.32	2.68 × 10^−2^	8.56 × 10^−3^
malQ ^a^	4-alpha-glucanotransferase	2.4.1.25	K00705	up	8.10	1.30 × 10^−9^	1.55 × 10^−8^
PYG ^b^	glycogen phosphorylase	2.4.1.1	K00688	up	4.89	3.68 × 10^−39^	2.75 × 10^−37^
GCK ^b^	glucokinase	2.7.1.2	K12407	up	5.25	1.10 × 10^−18^	2.73 × 10^−17^
GPI ^b^	glucose-6-phosphate isomerase	5.3.1.9	K01810	up	3.59	1.27 × 10^−7^	1.19 × 10^−6^
Pgm ^b^	phosphoglucomutase	5.4.2.2	K01835	down	−1.75	2.50 × 10^−2^	8.10 × 10^−3^
Cdc25 ^c^	Ras nucleotide exchange factor Cdc25	ND	K03099	up	3.19	2.01 × 10^−3^	8.97 × 10^−3^
Cyr1 ^c^	adenylate cyclase	4.6.1.1	K01768	up	7.29	4.54 × 10^−6^	3.39 × 10^−5^
PI3K ^c^	phosphatidylinositol-4,5-bisphosphate 3-kinase	ND	K00922	down	−3.31	4.44 × 10^−3^	1.82 × 10^−2^
PDK1 ^c^	3-phosphoinositide dependent protein kinase-1	2.7.11.1	K06276	up	2.04	1.91 × 10^−3^	8.55 × 10^−3^
SGK1 ^c^	serum/glucocorticoid-regulated kinase 1	2.7.11.1	K13302	up	3.27	2.40 × 10^−6^	1.90 × 10^−5^
AKT ^c^	RAC serine/threonine-protein kinase	2.7.11.1	K04456	up	1.56	1.32 × 10^−2^	4.70 × 10^−2^
CDK1 ^c^	cyclin-dependent kinase 1	2.7.11.22	K02087	up	8.13	7.43 × 10^−10^	9.12 × 10^−9^
CDK2 ^c^	cyclin-dependent kinase 2	2.7.11.22	K02206	up	6.58	6.67 × 10^−4^	3.28 × 10^−3^
CDK4 ^c^	cyclin-dependent kinase 4	2.7.11.22	K02089	up	11.44	1.27 × 10^−3^	5.83 × 10^−3^
CCNA ^c^	cyclin-A	ND	K06627	up	7.74	7.04 × 10^−8^	6.83 × 10^−7^
CCNE ^c^	G1/S-specific cyclin-E1	ND	K06626	up	2.40	4.23 × 10^−6^	3.22 × 10^−5^
AMPK ^d^	5’-AMP-activated protein kinase	2.7.11.11	K07198	down	−3.25	1.11 × 10^−12^	1.76 × 10^−11^
ATG1 ^d^	Autophagy-specific gene 1	2.7.11.1	K21357	down	−4.59	3.38 × 10^−28^	1.45 × 10^−26^

^a^ The genes are involved in the hydrolyzation of starch; ^b^ glycolysis; ^c^ cell cycles; or ^d^ autophagy of cells.

## Data Availability

The raw data supporting the conclusions of this article will be made available by the authors, without undue reservation.

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
