# Peer review of "Comparative Genomic and Transcriptomic Profiling Revealed the Molecular Basis of Starch Promoting the Growth and Proliferation of Balantioides coli"

_animals, 2023, doi:10.3390/ani13101608_

Round 1

Reviewer 1 Report

This study was conducted to assess the mechanism of starch on the growth and reproduction of B. coli. The work is interesting, however, there are some major comments to be addressed.

1-      Introduction should be extended to include the epidemiological pattern and profile of Balantioides coli and the clinical impact of the infection in human and animals.

2-      The full term should be mention for the first appearance as full  name in the manuscript then abbreviated. This should be done in the entire manuscript.

3-      The gap in research question should be more elaborated in the Introduction section. 

4-   Methods; what about including other strains for the parasite?

5- Discussion should be elaborated on the basis of your findings with explanation of the main reasons and please consider addition of limitations of the study at the end of discussion section.

6- Conclusions should be elaborated and please consider addition of some future suggestions in light of your findings.

Author Response

审稿人#1:

要点 1: 导言应扩大到包括大肠杆菌的流行病学模式和概况以及感染对人类和动物的临床影响。

回应1:感谢您的好建议。我们在清洁修订的MS的介绍部分(第53-66行)中补充了寄生虫的流行病学模式和临床影响的相关内容。

第2:在稿件中首次出现时应提及全名,然后缩写。这应该在整个手稿中完成。

回应2:感谢您的建议。我们在干净修订的MS中仔细检查并修改了这些问题。

第3点研究问题的差距应该在引言部分更详细地阐述。

回应3:感谢您的评论。我们在介绍干净修订的 MS(第 72-73 行)时补充了相关信息。

第4点方法;包括寄生虫的其他菌株怎么样?

回应 4: 本研究将大肠杆菌P011菌株培养并维持在改良的DMEM培养基中,用于转录组学和比较基因组分析,未纳入其他菌株。

第5点讨论应根据您的发现进行详细阐述,并解释主要原因,请考虑在讨论部分末尾添加研究的局限性。

回应 5: 感谢您的评论。我们在讨论部分的末尾添加了一段,以考虑增加研究的局限性 干净修订的MS.(第 514-518 行)

第6点应详细阐述结论,并考虑根据自己的发现增加一些未来的建议。

回应6: 感谢您的意见和建议。我们重写了结论部分,并考虑添加一些与我们的研究相关的未来建议,干净修订的MS。 (第 527-534 行)

Reviewer 2 Report

Dear Authors 

Your paper is of high scientific merit and very well performed.

Experimental techniques are also well carried out and presented . Only, please, 

write all the species names, as "in vitro" definition, in Italics a

Author Response

REVIEWER #2:

Point 1: Your paper is of high scientific merit and very well performed. Experimental techniques are also well carried out and presented. Only, please, write all the species names, as "in vitro" definition, in Italics a.

Response 1: We are so grateful to the reviewer for his positive comments on the MS. For these issues, we have carefully checked and modified one by one in the clean revised MS.

Reviewer 3 Report

The manuscript “Comparative genomic and transcriptomic profiling of Balantioides coli reveal the essential role of starch in its growth and proliferation”, describes a transcriptomic analysis of B. Coli grown in media supplemented or not with starch. In my opinion, the manuscript needs to be significantly improved in several aspects: rationale, structure and writing, but I do not see why it cannot be accepted for publication if those points are correctly addressed. In the following I depict my comments:

General comment: The manuscript would greatly benefit from an English language revision. In general, is comprehensible, but it can be certainly improved.

Line 1. Title: It is unclear if the statement made here, concerning the essential role of starch in the growth and proliferation, is correct. According to the sentence in Lines 58 and 59 Barbosa et al. already used starch to maintain in vitro for more than 2 years B. coli. If so, the essential role of starch in their growth and proliferation was already stated and published and the title should be amended. Actually, what the authors analysed was how B. coli used starch to grow and proliferate, by analysing the metabolic pathways used by the trophozoite.

Line 102. Materials and Methods. The authors stated that the N group did not received any starch supplementation in the media, but It is not clear from this sentence if they had any alternative source of carbon. Do this fact may explain the lack of growth observed in the results section? Do the authors think that this fact may affect the transcriptomics comparisons? How representative are N groups in that regard? Please modify discussion section according to the representativity expected.

Line 106. Materials and Methods. The procedure used to count and calculate the trophozoites densities should be explained in detail, not using a reference.

Lines 94 to 115. Materials and Methods. In these two paragraphs the authors describe how trophozoites were selected, cultured and the three groups used in the experiment. In general it is very hard to follow, specially I found very much confusing the description of the IS subgroup after the other two. I understand this third group was used as negative control in the transcriptomic analysis, but it is hard to understand in the way it is described. I strongly recommend the authors to rephrase both paragraphs for clarity using short, direct sentences.

Line 219. Figure 1. Considering that there were three replicates per group, there is no need to show the data as a box-plot, they may be represented as individual points. Also, it is unclear for me why the authors use “density” in the y-axis, I think “concentration” or “count” would be more correct; concentration/count refers to how much of a substance is present in a mixture, whereas density refers to the mass of a substance per unit volume. When the authors describe the procedures in more detail (Comment about Line 106) please refer to this point.

Author Response

REVIEWER #3:

General comments: General comment: The manuscript would greatly benefit from an English language revision. In general, is comprehensible, but it can be certainly improved.

Response: Thanks for your comments and suggestions. We have tried to avoid any grammar or syntax error, according to the comments, and asked a native English-speaking colleague who are skilled authors of English language papers to check the English, and overall style by the Language Editing Services of Letpub (https://www.letpub.com/) before submission.

 Point 1: Line 1. Title: It is unclear if the statement made here, concerning the essential role of starch in the growth and proliferation, is correct. According to the sentence in Lines 58 and 59 Barbosa et al. already used starch to maintain in vitro for more than 2 years B. coli. If so, the essential role of starch in their growth and proliferation was already stated and published and the title should be amended. Actually, what the authors analysed was how B. coli used starch to grow and proliferate, by analysing the metabolic pathways used by the trophozoite.

Response 1: Thanks for your reasonable comments. So we consider changing the title to “Comparative genomic and transcriptomic profiling of Balantioides coli reveal the molecular mechanism of starch promoting its growth and proliferation”.

 Point 2: Line 102. Materials and Methods. The authors stated that the N group did not receive any starch supplementation in the media, but It is not clear from this sentence if they had any alternative source of carbon. Do this fact may explain the lack of growth observed in the results section? Do the authors think that this fact may affect the transcriptomics comparisons? How representative are N groups in that regard? Please modify discussion section according to the representativity expected.

Response2: Thanks for your comment and suggestions. In fact, the basic DMEM medium originally contained glucose, but the basic media used in the N and T group of this study were same except for the presence of starch in the T group. In addition, the result of a previous experiment in the authors’ lab showed that the addition of glucose to the basic DMEM medium did not help the growth and development of B. coli, and the trophozoites would disappear after a period of time. Therefore, we think that the glucose in the basic DMEM medium can’t affect the transcriptomic comparisons, and the results of phenotypic and transcriptomic analysis in both N group and T group were representative.

 Point 3: Line 106. Materials and Methods. The procedure used to count and calculate the trophozoites densities should be explained in detail, not using a reference.

Response 3: Thanks for your comments. We have presented the detailed procedure for counting and calculating the trophozoite concentrations in the section “materials and methods” of the clean revised MS. (Line 120-127)

Point 4: Lines 94 to 115. Materials and Methods. In these two paragraphs the authors describe how trophozoites were selected, cultured and the three groups used in the experiment. In general, it is very hard to follow, specially I found very much confusing the description of the IS subgroup after the other two. I understand this third group was used as negative control in the transcriptomic analysis, but it is hard to understand in the way it is described. I strongly recommend the authors to rephrase both paragraphs for clarity using short, direct sentences.

Response 4: Thanks for your comments. We have modified the description and added a flow chart (Figure S1) to make the manuscript more complete and clear.

Point 5: Line 219. Figure 1. Considering that there were three replicates per group, there is no need to show the data as a box-plot, they may be represented as individual points. Also, it is unclear for me why the authors use “density” in the y-axis, I think “concentration” or “count” would be more correct; concentration/count refers to how much of a substance is present in a mixture, whereas density refers to the mass of a substance per unit volume. When the authors describe the procedures in more detail (Comment about Line 106) please refer to this point.

Response 5: Thanks for your comments. We have replotted Figure 1 and replaced the box plot with a line chart of average trophozoite concentration. Meanwhile, according to your suggestions, we changed the “density” to “concentration” in the plot and the clean revised MS. 

Reviewer 4 Report

The manuscript submitted for review presents as the main objective to perform both transcriptomic profiling based on single-cell RNA sequencing and comparative genomic analysis in order to reveal the molecular mechanism of the effect of starch on the proliferation of B. coli in vitro. No articles on the genomic structure and transcriptomic profiling of B. coli have been previously reported in the literature. Therefore the manuscript presented is highly original in this sense, as the manuscript submitted provides an interesting approach trying to associate the molecular mechanism as related to amylose metabolism to discover the metabolic pathways in B. coli by promoting the cell cycle and suppressing the autophagy of B. coli trophozoites proliferating in vitro, to the possible presentation and clinical severity of diarrheal disease (Balantidiasis) in weaned piglets. Overall the manuscript is well written and most of the appropriate citations have been included in the references section. However, the major question for this study is related to the experimental design and the number of cells and biological replicates used in the Single-cell RNA-seq analysis.

For example, If authors are looking for expression changes in B. coli trophozoite populations subject to distinct treatment categories (Starch vs No Starch added to DMEM culture medium) the number of cell samples utilized (10 B. coli trophozoites) is it enough for a well controlled and robust in vitro single-cell RNA seq experiment?. How many biological replicates per group might be sufficient to analyze how the culture composition of trophozoites changes with the starch treatment for a basic differential expression analysis? In other published RNA-seq studies researchers utilize at least 3 biological replicates per treatment/group, independent of the number of cells present per treatment/group. This, particularly in the case of an experiment in which no viability test assays were performed to assess if trophozoites were alive or morphologically intact, especially those cultured in the absence of Starch for 72 hours, which might have an impact on the overall downregulations of genes, as compared to the trophozoites cultured in the starch enriched medium where a high number of upregulated genes were found.

In addition, other points found by the reviewer and authors are invited to correct are the following:

Title

Please check if the scientific name “Balantioides coli” is correct. According to the Center for Diseases Control in the USA, while recent molecular analyses have suggested the need for taxonomic revision, and the parasite is now sometimes referred to as Neobalantidium coli or Balantioides coli, this nomenclature has neither been resolved nor widely adopted in the medical community. www.cdc.gov/parasites/balantidium/biology.html

Simple summary

Line 20. Please check if the term “Balantiosis” is correct and/or of current medical use. According to the Center for Diseases Control in the USA, when referring to the disease caused by B. coli “Balantidiasis” (also known as Balantidium coli infection) is used. CDC - Balantidiasis

Line 23. Can use “glucosa moities” instead of “glucoses”.

Line 24. Correct the term “arasite”

Line 25. Can use “balantidial diarrhea” instead of “balantial diarrhea”

Abstract

Line 30. Can use “replication” instead of “reproduction”

 Line 33. Correct the term “expended”

Line 35. Use “replication” instead of “reproduction”

 1.       Introduction

Lines 75-78. Use italics for “Toxoplasma gondii”, “T. gondii”, and “Cryptosporidium”. In addition check for the correct references cited as [10,11], as such citations refer to the transcriptome analysis of Tetrahymena thermophila, and the transcriptome expressed by Arthrobotrys oligosporaMonacrosporium cionopagum and Arthrobotrys dactyloides; respectively.

2.       Materials and Methods

2.1 Culture of Balantioides coli in vitro and experimental design

Lines 106-107. While it is clearly mentioned thatThree replicates of each group were performed to verify the reliability of the results” in the experiment in which trophozoites were cultured in the modified DMEM with or without starch (T and N groups, respectively), it is not clearly indicated if three biological replicates were also used for RNA extractions in order to conduct the transcriptomics studies. In line 112 it is mentioned that “Ten cells of each subgroup were placed in less than 0.5 μL of RNase-free” for RNA extraction, reverse-transcription cDNA amplification, Library construction and single-cell RNA sequencing.

Line 112. Please indicate what volumen authors refer to in expression “less than 0.5 μL of RNase-free water”.

2.5   Validation of DEGs by RT-qPCR

Lines 163-164. Please indicate why only 8 differentially expressed genes were selected for real time quantitative PCR, in order to validate the reliability of the data obtained by single cell RNA-seq.

 Lines 186-188. Use italics for all scientific names

Line 193. Correct the term “expended”

3. Results

 Line 211-211. In regard to video S1, Was a viability test study performed? Some of the trophozoites presented in video 1 appear static, and with apparent morphological damage. Are they dead trophozoites? What was the morphology and vitality of trophozoites unexposed to Starch?

 Line 226. Legend to Figure 1. Separate “inoculationwas”

3.2 Preliminary analysis of transcriptomic sequencing data

Please indicate the total number of clean reads obtained for each subgroup of analyzed trophozoites. Please indicate also how many biological replicates were single-cell RNA sequenced. Were samples containing 10 B. coli trophozoites each utilized for the IS, TS, and NS subgroup analyzed? Did the authors use three biological replicates or 3 technical replicates for each subgroup in the single-cell RNA seq transcriptomic analysis?

It is mentioned that different numbers of differentially expressed genes were found in the study, depending upon the subgroup comparison sets. For example, a total of 8876 DEGs were found in the NS vs IS set, most of which were downregulated. In the TS vs IS set, a total of 6493 DEGs were found in the TS subgroup; whereas for the TS vs NS set, 2231 downregulated genes and 5508 upregulated genes were identified in the TS subgroup, which were involved in various metabolic changes (Figure 3A).

Authors are invited to elaborate a little on this issue, particularly in the discussion section trying to relate the downregulation of genes in the NS group due to the possible effect of cell death in the trophozoites that were cultivated in the absence of starch. As compared to trophozoites that were not cultivated under the same conditions (IS group)?

Line 273. Figure 3. Drawings (Charts) are too small, and GO terms identifiers are very difficult to read.

 Line 371. Separate and place in italics “B.coli”.

Discussion

Line 403. The statement “Some previous data suggest that B. coli may be a facultative anaerobic organism.” Needs a reference

Line 408. Place “Saccharomyces cerevisiae” in italics

Line 416. Idem for “S. cerevisiae”

Line 428. Idem for “Tetrahymena”

References

Line 547. The correct citation for author names is: “García-Rodríguez JJ, Köster PC, Ponce-Gordo F.

Line 566. Correct species name and place “Caenorhabditis Briggsae” in italics.

Line 575. “Peritrichia” in italics

Line 578. Italics for “Arapaima gigas”

Lines 581 and 582. Italics in “Saccharomyces cerevisiae”.

Line 587. “Cryptococcus neoformans” in italics

Author Response

REVIEWER #4:

General comments: The manuscript submitted for review presents as the main objective to perform both transcriptomic profiling based on single-cell RNA sequencing and comparative genomic analysis in order to reveal the molecular mechanism of the effect of starch on the proliferation of B. coli in vitro. No articles on the genomic structure and transcriptomic profiling of B. coli have been previously reported in the literature. Therefore, the manuscript presented is highly original in this sense, as the manuscript submitted provides an interesting approach trying to associate the molecular mechanism as related to amylose metabolism to discover the metabolic pathways in B. coli by promoting the cell cycle and suppressing the autophagy of B. coli trophozoites proliferating in vitro, to the possible presentation and clinical severity of diarrheal disease (Balantidiasis) in weaned piglets. Overall the manuscript is well written and most of the appropriate citations have been included in the references section.

Response: Thanks for your positive comments. We have carefully replied point by point according to the review comments.

Point 1: However, the major question for this study is related to the experimental design and the number of cells and biological replicates used in the Single-cell RNA-seq analysis. For example, if authors are looking for expression changes in B. coli trophozoite populations subject to distinct treatment categories (Starch vs No Starch added to DMEM culture medium), the number of cell samples utilized (10 B. coli trophozoites) is it enough for a well controlled and robust in vitro single-cell RNA seq experiment? How many biological replicates per group might be sufficient to analyze how the culture composition of trophozoites changes with the starch treatment for a basic differential expression analysis? In other published RNA-seq studies researchers utilize at least 3 biological replicates per treatment/group, independent of the number of cells present per treatment/group. This, particularly in the case of an experiment in which no viability test assays were performed to assess if trophozoites were alive or morphologically intact, especially those cultured in the absence of Starch for 72 hours, which might have an impact on the overall downregulations of genes, as compared to the trophozoites cultured in the starch enriched medium where a high number of upregulated genes were found.

Response 1: Thanks for your comments. In the previously published work (Reference 12, Jiang CQ.et al), single-cell RNA-Seq method (SMART-seq2) was applied to about 2-20 cells of each group without biological replicate, whereas same technology was applied to 10 trophozoites of each subgroup in our work. We have tried to avoid the caused result by noise to the maximum extent and meet the demand of subsequent sequencing, we think it is enough for a well-controlled and robust in vitro single-cell RNA seq experiment. In fact, we designed three biological replicates each subgroup at the beginning, but some of the samples failed to be sequenced. Finally, we had to use one biological replicate with 10 trophozoites (cells) each subgroup to perform downstream analyses (Table S2). Although one replicate is really a pity in this experiment, we think our results are representative. The reasons are as follows: 1) One previous study has proved that SMART-seq2 transcriptome libraries have improved detection, coverage, bias and accuracy compared to traditional libraries (Simone Picelli. et al., 2013, SMART-seq2 for sensitive full-length transcriptome profiling in single cells, Nat Methods.); 2) Six common and prominent scRNA-seq methods were systematically compared based on sequencing data of one replicate in the previous study (Ziegenhain C. et al., 2017, Comparative Analysis of Single-Cell RNA Sequencing Methods, Mol Cell.), SMART-seq2 performed best than other methods from its highest sensitivity and accuracy and low amplification noise. Therefore, it is certainly that this technical advantage compensates for the absence of biological replicates.

Title

Point 2: Please check if the scientific name “Balantioides coli” is correct. According to the Center for Diseases Control in the USA, while recent molecular analyses have suggested the need for taxonomic revision, and the parasite is now sometimes referred to as Neobalantidium coli or Balantioides coli, this nomenclature has neither been resolved nor widely adopted in the medical community. www.cdc.gov/parasites/Bala- nti dium/biology.html.

Response 2: Thanks for your comments. As you said, the naming of Balantioides coli was once controversial, but we referred to the latest literature. Mathison and Pritt corrected their earlier taxonomic update and provided a taxonomic history of this species. For clinical reporting, the currently accepted name of this species is Balantioides coli and the clinical disease is balantiosis (Mathison BA. et al., 2021, Medical Parasitology Taxonomy Update, January 2018 to May 2020, J Clin Microbiol.). There is an increasing use of the correct genus name in recent publications and in up-to-date databases (i.e. in GenBank, all records of Balantidium coli / Neobalantidium coli are registered as Balantioides coli as the source organism) (Ponce-Gordo F. et al., 2021, Balantioides coli. Res Vet Sci.).

Simple summary

Point 3: Line 20. Please check if the term “Balantiosis” is correct and/or of current medical use. According to the Center for Diseases Control in the USA, when referring to the disease caused by B. coli “Balantidiasis” (also known as Balantidium coli infection) is used. CDC – Balantidiasis

Response 3: Thanks for your comments. Same as the Point 2, Mathison and Pritt pointed out that the currently accepted name of this species is Balantioides coli and the clinical disease is balantiosis (Mathison BA, Bradbury RS, Pritt BS. Medical Parasitology Taxonomy Update, January 2018 to May 2020. J Clin Microbiol. 2021 Jan 21;59(2):e01308-20. doi: 10.1128/JCM.01308-20).

Point 4: Line 23. Can use “glucosa moities” instead of “glucoses”.

Response 4: Thanks for your suggestion. We have changed the word “glucoses” to “glucosa moities” in the corresponding position the clean revised MS. (Line 26)

Point 5: Line 24. Correct the term “arasite”

Response 5: Thanks for your comment. We have changed the term “arasite” to “B. coli” in the corresponding position the clean revised MS. (Line 27)

Point 6: Line 25. Can use “balantidial diarrhea” instead of “balantial diarrhea”

Response 6: Thanks for your suggestion. We have changed “balantial diarrhea” to “balantidial diarrhea” in the clean revised MS.

Abstract

Point 7: Line 30. Can use “replication” instead of “reproduction”

Response 7: Thanks for your suggestion. We have changed “reproduction” to “replication” in abstract of the clean revised MS. (Line 33)

Point 8: Line 33. Correct the term “expended”

Response 8: Thanks for your comment. We have changed the word “expended” to “expanded” in the corresponding position the clean revised MS. (Line 36)

Point 9: Line 35. Use “replication” instead of “reproduction”

Response 8: Thanks for your suggestion. We have changed “reproduction” to “replication” in abstract of the clean revised MS. (Line 38)

  1. Introduction

Point 10: Lines 75-78. Use italics for “Toxoplasma gondii”, “T. gondii”, and “Cryptosporidium”. In addition check for the correct references cited as [10,11], as such citations refer to the transcriptome analysis of Tetrahymena thermophila, and the transcriptome expressed by Arthrobotrys oligospora, Monacrosporium cionopagum and Arthrobotrys dactyloides; respectively.

Response 10: Thanks for your comments. We are very sorry for the citation error here, we have revised it in the clean revised MS.

  1. Materials and Methods

2.1 Culture of Balantioides coli in vitro and experimental design

Point 11: Lines 106-107. While it is clearly mentioned that “Three replicates of each group were performed to verify the reliability of the results” in the experiment in which trophozoites were cultured in the modified DMEM with or without starch (T and N groups, respectively), it is not clearly indicated if three biological replicates were also used for RNA extractions in order to conduct the transcriptomics studies. In line 112 it is mentioned that “Ten cells of each subgroup were placed in less than 0.5 μL of RNase-free” for RNA extraction, reverse-transcription cDNA amplification, Library construction and single-cell RNA sequencing.

Response 11: Thanks for your comments. In fact, we designed three replicates of each subgroup to RNA extractions at the beginning, but some of the samples were discarded due to sequencing failure or low sequencing quality. Finally, we had to use one replicate for each subgroup to perform downstream analyses.

Point 12: Please indicate what volume authors refer to in expression “less than 0.5 μL of RNase-free water”.

Response 12: Thanks for your comments. What we want to express here is that we should carry as little RNase-free water as possible when separating single cells, and there is no specific volume value. We have revised this sentence in the clean revised MS. (Line 139-141)

2.5 Validation of DEGs by RT-qPCR

Point 13: Lines 163-164. Please indicate why only 8 differentially expressed genes were selected for real time quantitative PCR, in order to validate the reliability of the data obtained by single cell RNA-seq.

Response 13: Thanks for your comments. We referred to the relevant literature (Chen G. et al., 2018, Transcriptomics Sequencing Provides Insights into Understanding the Mechanism of Grass Carp Reovirus Infection, Int J Mol Sci.), most of them selected about 8 genes to verify, and 8 differentially expressed genes in our study were not randomly selected, but were important genes obtained according to Figure 9.

Point 14: Lines 186-188. Use italics for all scientific names

Response 14: Thanks for your comments and suggestions. We have examined one by one carefully and have modified in the clean revised MS.

Point 15: Line 193. Correct the term “expended”

Response 15: Thanks for your comment. We have changed the word “expended” to “expanded” in the corresponding position in the clean revised MS. (Line 221)

3.Results

Point 16: Line 211-211. In regard to video S1, Was a viability test study performed? Some of the trophozoites presented in video 1 appear static, and with apparent morphological damage. Are they dead trophozoites? What was the morphology and vitality of trophozoites unexposed to Starch?

Response 16: Thanks for your questions. The video S1 was recorded when we found that trophozoites grew well in starch-containing medium during culture. As shown in this video, they are not dead trophozoites, but debris and granules of starch. According to our experimental observations (6, Yan W.et al), in the medium without starch, the trophozoites move slowly or static, no starch granules can be seen in the trophozoites and the number of trophozoites is distinctly lower than those in the medium with 5 mg/mL.

Point 17: Line 226. Legend to Figure 1. Separate “inoculationwas”

Response 17: Thanks for your comment. We are sorry that a space is missing here. We have modified it in the clean revised MS. (Line 255)

3.2 Preliminary analysis of transcriptomic sequencing data

Point 18: Please indicate the total number of clean reads obtained for each subgroup of analyzed trophozoites. Please indicate also how many biological replicates were single-cell RNA sequenced. Were samples containing 10 B. coli trophozoites each utilized for the IS, TS, and NS subgroup analyzed? Did the authors use three biological replicates or 3 technical replicates for each subgroup in the single-cell RNA seq transcriptomic analysis?

Response 18: Thanks for your comments. The total number of raw and clean reads obtained for each subgroup were presented in Table S2. Yes, only one biological replicate containing 10 B. coli trophozoites for each subgroup were utilized for the single-cell RNA-sequencing. Although one replicate is really a pity in this experiment, we think our results are representative. The reasons are as follows: 1) One previous study has proved that SMART-seq2 transcriptome libraries have improved detection, coverage, bias and accuracy compared to traditional libraries (Simone Picelli. et al., 2013, SMART-seq2 for sensitive full-length transcriptome profiling in single cells, Nat Methods.); 2) Six common and prominent scRNA-seq methods were systematically compared based on sequencing data of one replicate in the previous study (Ziegenhain C. et al., 2017, Comparative Analysis of Single-Cell RNA Sequencing Methods, Mol Cell.), SMART-seq2 performed best than other methods from its highest sensitivity and accuracy and low amplification noise. Therefore, it is certainly the most suitable method when annotating and/or quantifying transcriptomes of fewer cells.

Point 19: It is mentioned that different numbers of differentially expressed genes were found in the study, depending upon the subgroup comparison sets. For example, a total of 8876 DEGs were found in the NS vs IS set, most of which were downregulated. In the TS vs IS set, a total of 6493 DEGs were found in the TS subgroup; whereas for the TS vs NS set, 2231 downregulated genes and 5508 upregulated genes were identified in the TS subgroup, which were involved in various metabolic changes (Figure 3A).

Response 19: Thanks for your comments. DEGs were identified in our study based on the three comparison subgroups. Here, we don’t descript the corresponding KEGG metabolic functions for the three groups, respectively. Instead, the total DEGs from them were utilized to quickly focus on the interested gene expression pattern of Cluster 4 via a clustering heatmap (Fig 4), and shown its metabolic changes in Fig 2B.    

Point 20: Authors are invited to elaborate a little on this issue, particularly in the discussion section trying to relate the downregulation of genes in the NS group due to the possible effect of cell death in the trophozoites that were cultivated in the absence of starch. As compared to trophozoites that were not cultivated under the same conditions (IS group)?

Response 20: Thanks for your comments and suggestions. In the NS subgroup, the picked trophozoites were not dead cells, and can still move slowly in the absence of starch at 72 h. Compared to trophozoites in the IS subgroup, the medium of NS subgroup did not contain starch, and the starch stored in the trophozoites was depleted in the later stage of culture, and could affect life activities of trophozoites. As shown in Figure S2A, in most GO terms, including CC, MF, and BP categories, the number of down-regulated genes was significantly higher than the number of up-regulated genes, this suggests that without starch, many life activities of B. coli are affected.

Point 21: Line 273. Figure 3. Drawings (Charts) are too small, and GO terms identifiers are very difficult to read.

Response 21: Thank for you comments. We have provided the updated Figure 3 with higher resolution to make it easy to observe.

Point 22: Line 371. Separate and place in italics “B.coli”.

Response 22: Thanks for your comments. We have modified it in the clean revised MS. (Line 399)

Discussion

Point 23: Line 403. The statement “Some previous data suggest that B. coli may be a facultative anaerobic organism.” Needs a reference

Response 23: Thanks for your comment. We have added the reference in the clean revised MS (Line 445).

Point 24: Line 408. Place “Saccharomyces cerevisiae” in italics

Response 24: Thanks for your comments. We have modified it in the clean revised MS. (Line 450)

Point 25: Line 416. Idem for “S. cerevisiae”

Response 25: Thanks for your comments. We have modified it in the clean revised MS. (Line 458)

Point 26: Line 428. Idem for “Tetrahymena”

Response 26: Thanks for your comments. We have modified it in the clean revised MS. (Line 470)

References

Point 27: Line 547. The correct citation for author names is: “García-Rodríguez JJ, Köster PC, Ponce-Gordo F.”

Response 27: Thanks for your comments. We have modified it in the clean revised MS. (Line 602)

Point 28: Line 566. Correct species name and place “Caenorhabditis Briggsae” in italics.

Response 28: Thanks for your comments. This literature was misquoted and we have deleted it.

Point 29: Line 575. “Peritrichia” in italics

Response 29: Thanks for your comments. We have modified it in the clean revised MS. (Line 634)

Point 30: Line 578. Italics for “Arapaima gigas”

Response 30: Thanks for your comments. We have modified it in the clean revised MS. (Line 637)

Point 31: Lines 581 and 582. Italics in “Saccharomyces cerevisiae”.

Response 31: Thanks for your comments. We have modified it in the clean revised MS. (Line 640 and 641)

Point 32: Line 587. “Cryptococcus neoformans” in italics

Response 32: Thanks for your comments. We have modified it in the clean revised MS. (Line 646)

Round 2

Reviewer 1 Report

Authors addressed the major raised comments. However, my main concern during the second round is about the number of replicates. This comment has been raised by other reviewers and I agree with their comment. Despite the explanation and the other supporting methodology provided by authors, one replicate is not sufficient to reflect general findings. Authors mentioned in their response to other reviewers that a previous study ‘’ Ziegenhain C. et al., 2017, Comparative Analysis of Single-Cell RNA Sequencing Methods, Mol Cell ‘’ was based on one replicate but I checked this comment and I noticed that the mentioned study was not based on one replicate. The methodology should be conducted correctly and should not be based on sporadic studies. I am also wondering that authors even did not include the fact that some of their samples failed to be sequenced in limitation section added. Clearly, without a proper methodology frame, extrapolating the result to a general finding is erroneous. Given these serious flaws, additional experiments needed (mainly number of replicates which is critical) and my suggestion is to reject the manuscript and encourage its resubmission after verification of the findings by more replicates.

Author Response

REVIEWER #1

Q1: Authors addressed the major raised comments. However, my main concern during the second round is about the number of replicates. This comment has been raised by other reviewers and I agree with their comment. Despite the explanation and the other supporting methodology provided by authors, one replicate is not sufficient to reflect general findings. Authors mentioned in their response to other reviewers that a previous study ‘’ Ziegenhain C. et al., 2017, Comparative Analysis of Single-Cell RNA Sequencing Methods, Mol Cell ‘’ was based on one replicate but I checked this comment and I noticed that the mentioned study was not based on one replicate. The methodology should be conducted correctly and should not be based on sporadic studies. I am also wondering that authors even did not include the fact that some of their samples failed to be sequenced in limitation section added. Clearly, without a proper methodology frame, extrapolating the result to a general finding is erroneous. Given these serious flaws, additional experiments needed (mainly number of replicates which is critical) and my suggestion is to reject the manuscript and encourage its resubmission after verification of the findings by more replicates.

Response to Q1: In this study, we designed three biological replicates each subgroup for single-cell RNA-seq at the beginning, but some of the samples failed to be sequenced. Finally, we had to use one biological replicate each subgroup to perform downstream analyses. Although one replicate is really a pity in this experiment, we think our results are reliable, not sporadic. The reasons are as follows: 1) The results of single-cell RNA-seq have been further verified by qRT-PCR based on three biological replicates of each subgroup; 2) The data of comparative genomics have also supported the results of transcriptomic analysis in this study; 3) One previous study has proved that SMART-seq2 transcriptome libraries have improved detection, coverage, bias and accuracy compared to traditional libraries (Simone Picelli. et al., 2013, SMART-seq2 for sensitive full-length transcriptome profiling in single cells, Nat Methods.), and therefore the advantage of SMART-seq2 will make the results of transcriptomic analysis of B. coli in our study are more accurate than routine transcriptomic analysis. In addition, we have accordingly supplemented the fact that some of samples failed to be sequenced in limitation section of the clean revised MS. (Line 500-501)

Reviewer 3 Report

The manuscript can be accepted in the present form after the changes introduced

Author Response

Thanks for your positive comments on the revised MS. We sincerely appreciate your support and assistance.

Reviewer 4 Report

The manuscript submitted for review presents as the main objective to perform both transcriptomic profiling based on single-cell RNA sequencing and comparative genomic analysis in order to reveal the possible molecular mechanisms involved on the effect of starch on the proliferation of Balantioides coli in vitro. No articles on the genomic structure and transcriptomic profiling of B. coli have been previously reported in the literature. Therefore, the manuscript presented is highly original in this sense, as the manuscript submitted provides an interesting approach trying to associate the molecular mechanism as related to amylose metabolism to discover the metabolic pathways found in B. coli by promoting the cell cycle and suppressing the autophagy of B. coli trophozoites proliferating in vitro, to the possible presentation and clinical severity of diarrheal disease (Balantidiasis) in weaned piglets. Overall the manuscript is well written and the appropriate citations have been included in the references section.

Author Response

(The authors gave the same response as above.)
